# Membrane bending occurs at all stages of clathrin-coat assembly and defines endocytic dynamics

Brandon L. Scott [1,2], Kem A. Sochacki[3], Shalini T. Low-Nam[1,2,6], Elizabeth M. Bailey[1,2], QuocAhn Luu[4,5], Amy Hor[4,5], Andrea M. Dickey[3], Steve Smith[4,5], Jason G. Kerkvliet[1,2], Justin W. Taraska [3] & Adam D. Hoppe [1,2]

Clathrin-mediated endocytosis (CME) internalizes plasma membrane by reshaping small regions of the cell surface into spherical vesicles. The key mechanistic question of how coat assembly produces membrane curvature has been studied with molecular and cellular structural biology approaches, without direct visualization of the process in living cells; resulting in two competing models for membrane bending. Here we use polarized total internal reflection fluorescence microscopy (pol-TIRF) combined with electron, atomic force, and super-resolution optical microscopy to measure membrane curvature during CME. Surprisingly, coat assembly accommodates membrane bending concurrent with or after the assembly of the clathrin lattice. Once curvature began, CME proceeded to scission with robust timing. Four color pol-TIRF showed that CALM accumulated at high levels during membrane bending, implicating its auxiliary role in curvature generation. We conclude that clathrin-coat assembly is versatile and that multiple membrane-bending trajectories likely reflect the energetics of coat assembly relative to competing forces.

---

[1] Department of Chemistry and Biochemistry, Avera Health and Sciences Building, RM 131 South Dakota State University (SDSU), Brookings, SD 57007, USA. [2] BioSNTR SDSU, Brookings, SD 57007, USA. [3] Laboratory of Molecular Biophysics, National Heart Lung and Blood Institute, National Institutes of Health, Bethesda, MD 20892, USA. [4] Nanoscience and Nanoengineering, South Dakota School of Mines and Technology (SDSMT), Rapid City, SD 57701, USA. [5] BioSNTR, SDSMT, Rapid City, SD 57701, USA. [6] Present address: Department of Chemistry, University of California, Berkeley, CA 94720, USA. Correspondence and requests for materials should be addressed to J.W.T. (email: justin.taraska@nih.gov) or to A.D.H. (email: adam.hoppe@sdstate.edu)

The clathrin coat, along with its auxiliary proteins, deforms plasma membrane into small ~100 nm endocytic vesicles, to mediate retrieval of membrane and membrane proteins from the cell surface[1,2]. This process has been intensely studied because of its importance in cell biology, and as a model of a dynamic supermacromolecular assembly. However, the progression of clathrin-coat assembly and its relationship to curvature have remained unclear, with morphologies inferred from static electron microscopy images and structural biology models[3–10]. Observations made by electron microscopy over 35 years ago suggested that clathrin oligomerizes into flat hexagonal lattices on the plasma membrane that rearrange into spheres via the transition of some of the lattice hexagons into pentagons[9]. However, energetic arguments suggest that this structural conversion would be energetically costly, requiring large protein contacts or ostensibly rigid structures to be broken and reformed during the hexagon to pentagon transition. This leads to a model in which membrane bending occurs progressively with the assembly of clathrin[7,8,11] (Fig. 1a). Recently, electron tomography/correlative fluorescence microscopy of thick-sections supported the model of pre-assembled flat clathrin sheets that subsequently bend into vesicles[10] (Fig. 1b). Thus, there remains uncertainty in the fundamental mechanism by which the membrane bends during clathrin-coat assembly.

Total internal reflection fluorescence (TIRF) microscopy has been an invaluable tool for analyzing the dynamics of endocytosis and associated proteins at the cell surface[6]. Axelrod and colleagues pioneered pol-TIRF microscopy to generate contrast between vertical and horizontal DiI-C18-labeled plasma membrane in living cells[12–16]. Using this approach, they were able to capture the dynamics of membrane topographical changes during exocytosis of chromaffin granules[11]. Here, we have refined pol-TIRF micoscopy and validated its ability to accurately measure the small changes in membrane topography by electron and atomic force microscopy (AFM). Computer simulations of two topographical membrane-bending models predicted the relationship between pol-TIRF, clathrin and dynamin signals. We demonstrate that in cells genome edited to express fluorescent protein-tagged clathrin and dynamin, clathrin-coated structures can form as curved membranes that accumulate clathrin or as pre-accumulated flat clathrin sites that then bend membrane to form pits. Both behaviors were observed within single living cells, suggesting that local biochemical and biophysical factors can mediate a switch between these two modes. This work demonstrates that the clathrin coat is flexible and that the rate-limiting step for vesicle formation is the induction of membrane bending, which may be regulated by axillary proteins such as CALM.

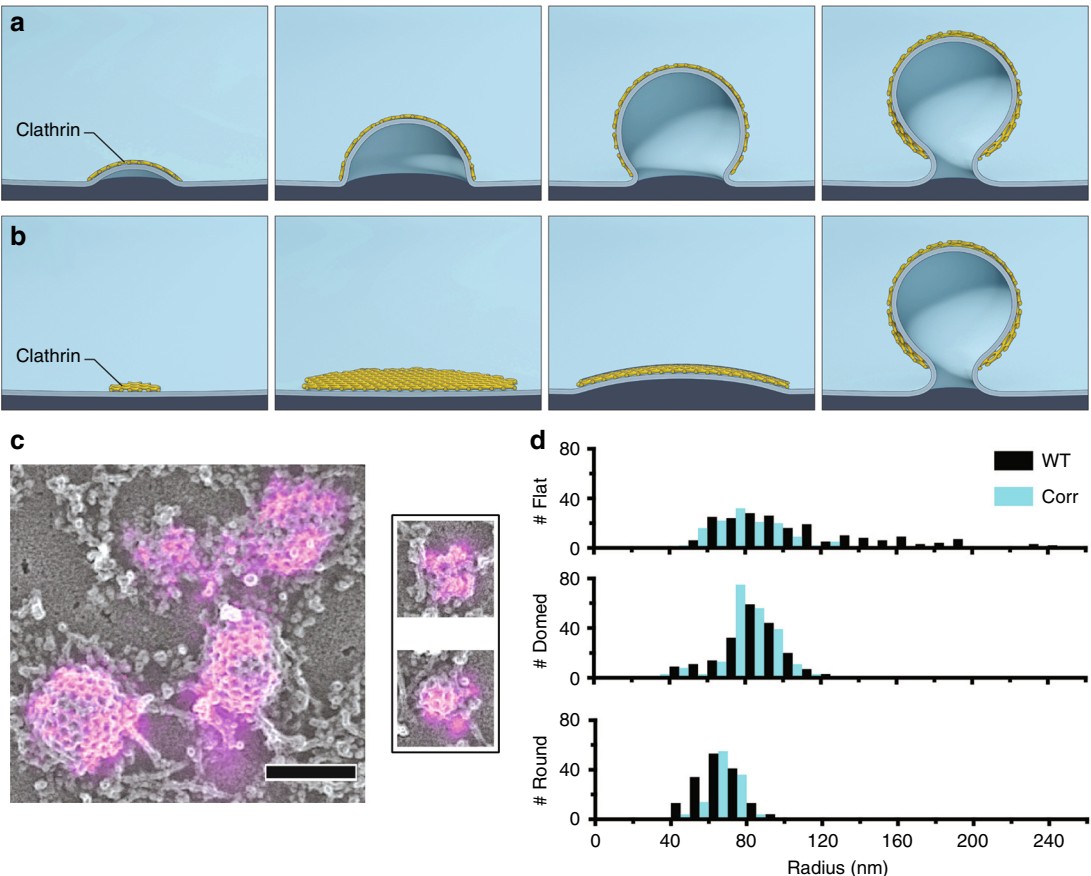

**Fig. 1** Mechanisms of membrane bending during CME and clathrin ultrastructure in unroofed SK-MEL-2 cells. **a** Schematic representation of CME in which membrane bending proceeds with a fixed radius of curvature during the addition of clathrin subunits. **b** Representation of CME in which clathrin first assembles into a flat sheet that remodels into a vesicle. **c** Correlative dSTORM and platinum-replica TEM images of fluorescently labeled clathrin (magenta) demonstrate a range of heterogeneous topographies even at the earliest stages of CME; scale bar is 200 nm. **d** The size distribution observed amongst clathrin structures in WT SK-MEL-2 cells (black) or SK-MEL-2 cells exogenously expressing clathrin light chain for correlative microscopy (blue). The inset shows ratios of average fluorescence associated with round vs. domed and domed vs. flat structures (SD shown for $N = 3$ cell membranes)

## Results

**CLEM imaging of clathrin assembly.** The two membrane-bending models exhibit distinct and opposing relationships between changing pit size and incorporation of new clathrin subunits. We applied single-molecule super-resolution immuno-fluorescence of clathrin light chain combined with platinum-replica correlative electron microscopy to image the structure of even the smallest clathrin-coated assemblies at the plasma membrane of SK-MEL-2 cells (Fig. 1c). The fluorescence signal of clathrin-coat proteins associated with clathrin structures increased from flat to domed to highly curved clathrin structures, suggesting that clathrin was added during these transitions (Fig. 1d). Additionally, the observed morphologies were heterogeneous and displayed a range of lateral radii (Fig. 1d and Supplementary Figures 3, 4), raising the possibility that clathrin accommodates multiple modes of membrane bending as well as the addition of new clathrin subunits at different morphological stages[17,18].

**Computer simulation of membrane-bending signals as measured by pol-TIRF.** Measuring the dynamics of membrane bending during clathrin assembly at single endocytic sites in living cells is necessary to distinguish the possible modes of membrane bending. Pol-TIRF has been used to image changes in membrane topography during exocytosis of chromaffin granules, which are much larger than CME sites[11,19,20]. We developed a microscope capable of creating pol-TIRF fields that were parallel (s-pol, S) or perpendicular (p-pol, P) to the coverslip with improved spatial uniformity by averaging multiple illumination directions[21,22] (Supplementary Figure 2). The P and S fields were used to selectively excite DiI molecules in vertical or horizontal membrane, respectively, thereby encoding membrane curvature into the ratio of P/S fluorescence images (Fig. 2a). A computer simulation of pol-TIRF for the formation of 100 nm vesicles by either model (Fig. 1a, b) predicted that the P/S image was sensitive to small changes in membrane bending (Fig. 2c). Although the simulation predicted small differences between the two CME

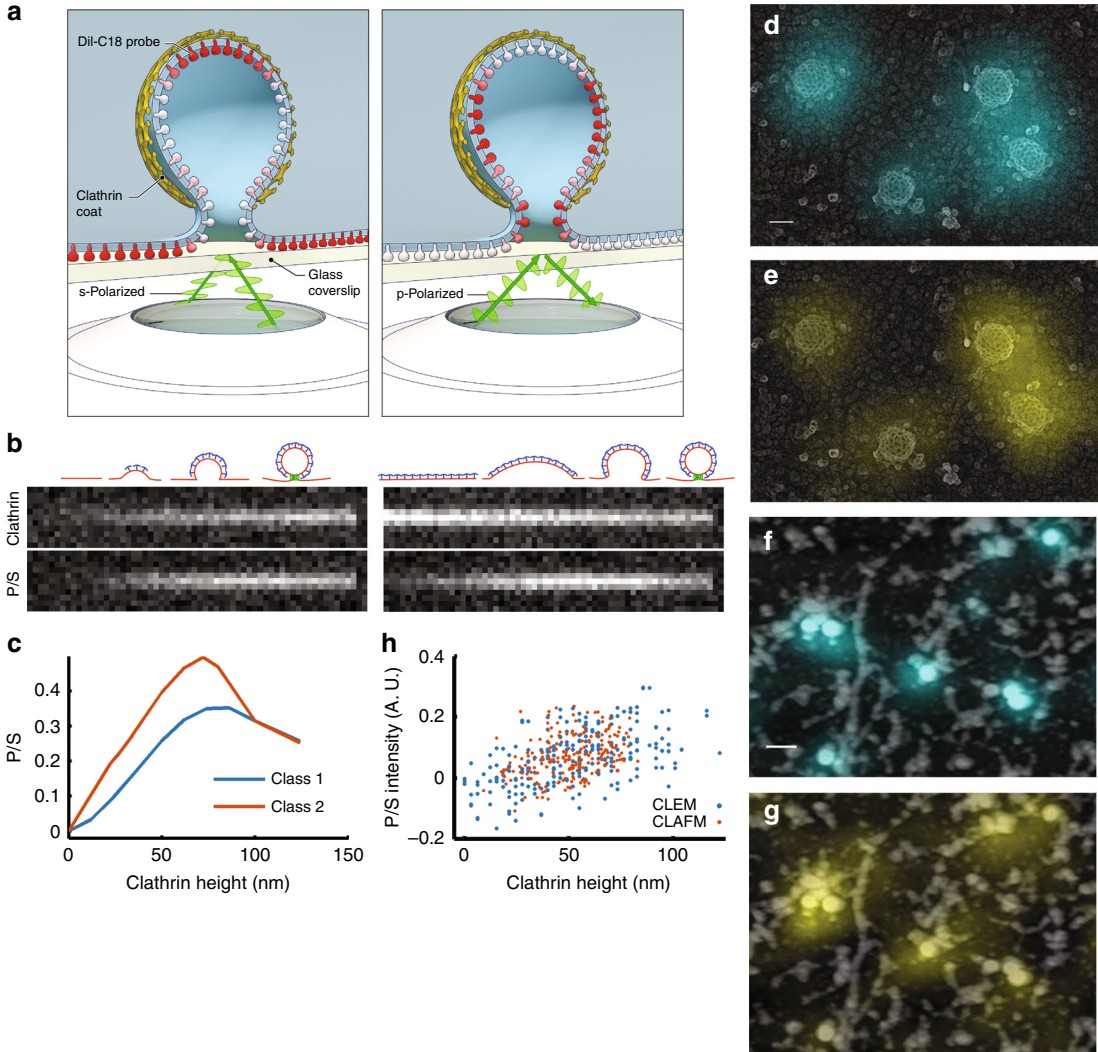

**Fig. 2** Polarized-TIRF microscopy enables imaging of membrane bending at clathrin-coated structures. **a** Schematic representation of pol-TIRF. DiI–C18 orients its dipole moment with the plasma membrane. S-polarized TIRF illuminates horizontal dye molecules, whereas P-polarized TIRF selectively excites vertical dye molecules. The P/S provides contrast for membrane bending. **b** Simulation of pol-TIRF signals for Class 1 and Class 2 membrane bending. **c** Quantification of high-resolution simulation at 10 discrete points for Class 1 (blue) and Class 2 (orange) in the absence of noise. **d** Correlative TEM-pol-TIRF imaging. Overlay of fluorescence from endogenous clathrin-Tq2 on the TEM micrograph showing four clathrin structures; scale bar is 100 nm. **e** Overlay of P/S on the same region of the micrograph. **f** Correlative AFM-pol-TIRF imaging; Overlay of clathrin-Tq2 on AFM micrograph; scale bar is 250 nm. **g** Overlay of P/S signal on the same region of the AFM micrograph. **h** Quantification of P/S intensity and heights from correlative tomographic reconstructions (blue), and AFM (orange)

models, these models can be readily distinguished by comparing P/S with the arrival of clathrin (Fig. 2b and Supplementary Figure 1). For membrane bending during assembly, clathrin and P/S increase together as the pit forms (Fig. 2b). Conversely, for the model in which a flat clathrin patch is reshaped into a sphere, the clathrin intensity is maximal prior to changes in P/S, and then decreases as bending moves the top of the structure deeper into the exponentially decaying TIRF field (Fig. 2b and Supplementary Figure 1). We considered the possibility that detection of the P/S signal would be less sensitive than detection of fluorescent clathrin arrival, thereby creating an apparent temporal delay between clathrin arrival and membrane bending. However, simulations of P/S and clathrin over a range of signal-to-noise ratios (SNRs) indicated that detection of P/S is weakly dependent on the SNR, and that the two models are distinguishable over a wide range of SNRs encountered in our live-cell imaging data (Fig. 2b and Supplementary Figure 1).

**Validation of pol-TIRF by correlative imaging**. Next, we directly validated pol-TIRF's sensitivity to membrane bending during CME by correlative light and electron (CLEM), and light and atomic force microscopy (CLAFM). In pol-TIRF-CLEM, endogenous clathrin-Tq2 fluorescence colocalized with the expected clathrin ultrastructure (Fig. 2d) and corresponding P/S signals were observed on individual clathrin-coated pits over a range of invagination stages (Fig. 2e). Importantly, P/S increased with pit stage as determined by morphology (Supplementary Figure 3), and with pit heights determined from TEM tomograms (Fig. 2h, $\rho = 0.548$, $p < 0.001$, and Supplementary Figure 4). Pol-TIRF-CLAFM on wet samples confirmed pol-TIRF's sensitivity for membrane bending, despite the reduced resolution achieved by AFM owing to the softness of biological samples. Specifically, the endogenous clathrin-Tq2 and P/S overlaid with peaks in the AFM images (Fig. 2f, g) and pit height and P/S were positively correlated (Fig. 2h, $\rho = 0.351$, $p < 0.001$, and Supplementary Figure 4). The observed variability in the P/S ratio relative to the CLEM and CLAFM measurements was predominantly a result of adjacent topographical features containing vertical membrane being located within the optical resolution of the measured clathrin structure (Fig. 2h and Supplementary Figure 4). Given this limitation, the pol-TIRF P/S ratio showed good agreement with both EM and AFM measurements and is sensitive to membrane bending on the length scale of CME.

**Live-cell imaging and tracking of membrane-bending dynamics during CME**. Given that pol-TIRF could reliably detect nanoscale changes in membrane bending, we recorded the dynamics of membrane bending at single endocytic events in SK-MEL-2 cells labeled with DiI that express gene-edited clathrin-Tq2 and dynamin2-eGFP[23] (Supplementary Figure 5 and Supplementary Movies 1–2). Single diffraction-limited endocytic events were tracked[24], filtered to retain only those that contained isolated clathrin, dynamin, and P/S events, and categorized based on detection of membrane bending relative to clathrin arrival (Fig. 3). The reliability of the detection of P/S relative to clathrin arrival can be seen in the example traces (Supplementary Figures 7–9). From these cells, ~7100 tracks had clathrin-Tq2 signatures that appeared and disappeared during the time of imaging. Of these tracks, 481 were selected for analysis based on a set of criteria, the most stringent of which were the absence of adjacent membrane curvature signals in the P/S image and a dynamin signature (Supplementary Figure 6a). In approximately half of the CME events, membrane bending was detected at the moment clathrin arrived and then grew in intensity (Fig. 3a, Class 1), consistent with bending during clathrin-coat assembly.

In the other half of the events, clathrin accumulated prior to detection of membrane bending (Fig. 3b, c). The delayed bending group was divided into two categories—a small subset in which all of the clathrin accumulated at the endocytic site prior to membrane bending (Fig. 3b, Class 2), and a larger group, in which some clathrin accumulated prior to bending, but additional clathrin was then added during the membrane bending (Fig. 3c, Class 3). The relative proportions of these events were Class 1 (43%), Class 2 (14%), and Class 3 (43%) (Fig. 3d). Thus, within the same cell, clathrin assembly and membrane bending occurs with heterogeneous timing and the clathrin coat accommodates several modes of membrane bending.

**Variable timing of membrane bending relative to clathrin assembly**. A population view of membrane-bending dynamics during CME revealed a variable delay between clathrin assembly and the onset of membrane bending. We observed that the clathrin lifetimes for Class 1 were shorter than Class 2 and 3 events (Class 1 ($80 \pm 42$ s), Class 2/3 ($108 \pm 39$ s), $p < 0.05$, Fig. 4a). Unlike clathrin, the lifetimes of membrane bending and dynamin showed no statistical differences across classes (Fig. 4a), indicating that the dynamics of these processes are identical regardless of class. Thus, Class 2/3 events had a delay in progression that was not present in Class 1 events. Delays of unknown mechanism have been suggested for CME[25] and a checkpoint related to membrane bending has been proposed[24]. In comparison with Class 1, Class 2/3 events lagged behind the start of clathrin assembly with $\Delta t = 25.5$ s (Fig. 4b). This same delay was also observed when comparing the lag for dynamin between Class 1 and Class 2/3 ($\Delta t = 24.1$ s) (Fig. 4b). We observed minimal differences between the dynamin lag from P/S initiation across the classes ($\Delta t = 0.2$ s, Fig. 4b), indicating that the principle difference in lifetime for the two classes arose during the time that clathrin began to assemble and the onset of membrane bending (Fig. 4c).

These results raise the possibility that the initial moments of clathrin association with the plasma membrane determine whether bending will begin immediately or if a flat intermediate state will form. Correlative dSTORM—platinum replica TEM of very-small clathrin structures containing only a few triskelia revealed both flat and curved morphologies (Fig. 1c), consistent with a bifurcation for entry into either Class 1 or Class 2/3 occurring early in the assembly process. Factors that influence this bifurcation could include the shape of the membrane at the moment clathrin binds, lateral membrane tension[26], curvature sensing/generating proteins[27], the stabilization of a curved state by factors such as AP2[28], or engagement of the actin cytoskeleton[25]. Consistent with a stabilization step being required, we observed many short-lived ($<18$ s) flat clathrin structures that did not recruit dynamin (Supplementary Figure 6b–d). Close inspection of the CME events did not reveal any examples of P/S signals that preceded clathrin, suggesting that either initial membrane topography was not a factor in defining the sites at which clathrin assembled or that the scale of membrane bending needed to recruit clathrin was below detection. Essentially, all CME structures that acquired a P/S signal acquired dynamin, indicating that once membrane bending starts, progression to a vesicle is robust (Fig. 4b). Thus, within the same cell, clathrin bends the membrane through multiple heterogeneous pathways in which the initiation of curvature is likely a rate-limiting step (Fig. 4c).

**Auxiliary protein CALM accumulates at high density during membrane bending**. CME involves the recruitment of many auxiliary proteins[6]. The protein CALM has been

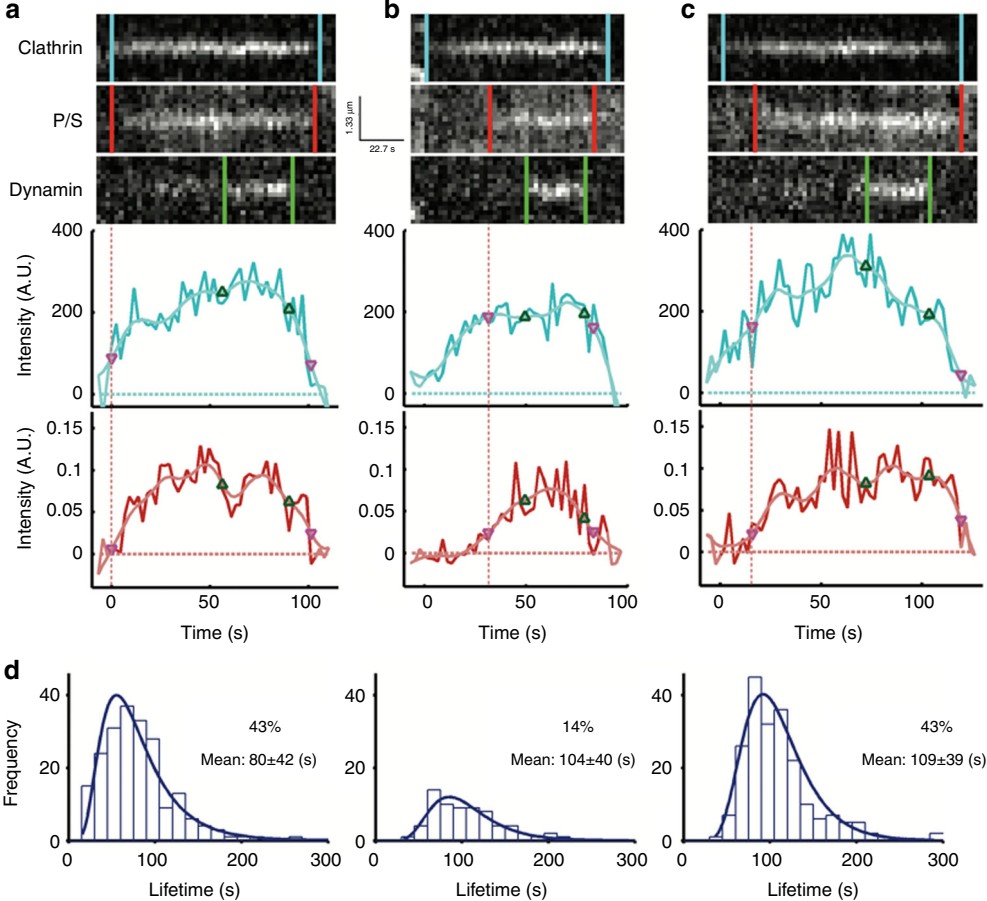

**Fig. 3** Distinct modes of membrane bending observed by pol-TIRF. **a–c** Kymographs of clathrin-Tq2, membrane bending (P/S), dynamin-eGFP, and corresponding intensity traces for clathrin (cyan) and P/S (red) for the three classes of membrane bending observed by pol-TIRF. The colored lines indicate the lifetime for each event, the dashed red line highlights the start of P/S event. **a** Class 1: clathrin and P/S signals proceed together indicating membrane bending as clathrin assembles. **b** Class 2: clathrin signal plateaus prior to the start of P/S indicating all required clathrin was present as a flat sheet. **c** Class 3: clathrin assembles prior to P/S signal, but new clathrin was recruited as the membrane bends and the vesicle is formed. **d** The clathrin lifetime histogram of each class from an average of four cells ($N = 481$ tracks total)

shown to facilitate membrane bending and regulate pit size by binding clathrin, AP2, and the plasma membrane to facilitate CME[29]. Additionally, correlative light/platinum replica TEM indicated that CALM associates at low levels with flat clathrin structures, but is greatly enriched on curved membranes[30], supporting the idea that CALM may dynamically regulate membrane curvature. To image CALM in the context of clathrin accumulation and membrane bending, we expanded the design of the microscope to include an additional imaging channel (Supplementary Figure 2) and expressed CALM-iRFP682 at low levels via a retroviral expression. Pol-TIRF showed that CALM was robustly recruited in a biphasic manner to sites of CME, arriving first at low density and then accumulating a higher density either before or during membrane bending (Fig. 5). CALM was robustly recruited to CME sites during membrane bending regardless of which membrane bending class (based on clathrin and P/S signals) was observed (Fig. 5). Thus, clathrin assembly can be coincident with membrane bending or precede membrane bending, but the arrival of CALM correlates strongly with the generation of curvature. This tight dynamic relationship along with extensive structural data[29] indicates that CALM likely contributes to the energy needed for membrane bending.

## Discussion

Here we have measured the membrane-bending dynamics during assembly of single-CME events using newly developed methods. These data provide the first direct observations of clathrin's ability to accommodate multiple membrane-bending trajectories that are not predicted by the current structural models of endocytosis (Fig. 1a, b). Rather, the observed curvature trajectories favor a model in which cooperating and competing energies define the trajectory. Factors influencing these energetics could include lateral membrane tension[26], local actin polymerization[25], cargo loading, cargo identity[31], chemical signals, charged and uncharged lipid sorting, and the recruitment of auxiliary CME components[32,33]. Multiple bending trajectories could be observed on the plasma membrane of single living cells. This suggests that the factors influencing membrane bending are restricted to small regions of the plasma membrane and could locally modulate the process. For example, local membrane tension differences or the subcellular distribution of auxiliary factors including CALM could regulate membrane bending[29]. Further work to elucidate the dynamics of other auxiliary CME proteins, membrane tension, and other signals during membrane bending, will be needed to develop a comprehensive understanding of the structural and biophysical nature of membrane curvature generation during CME.

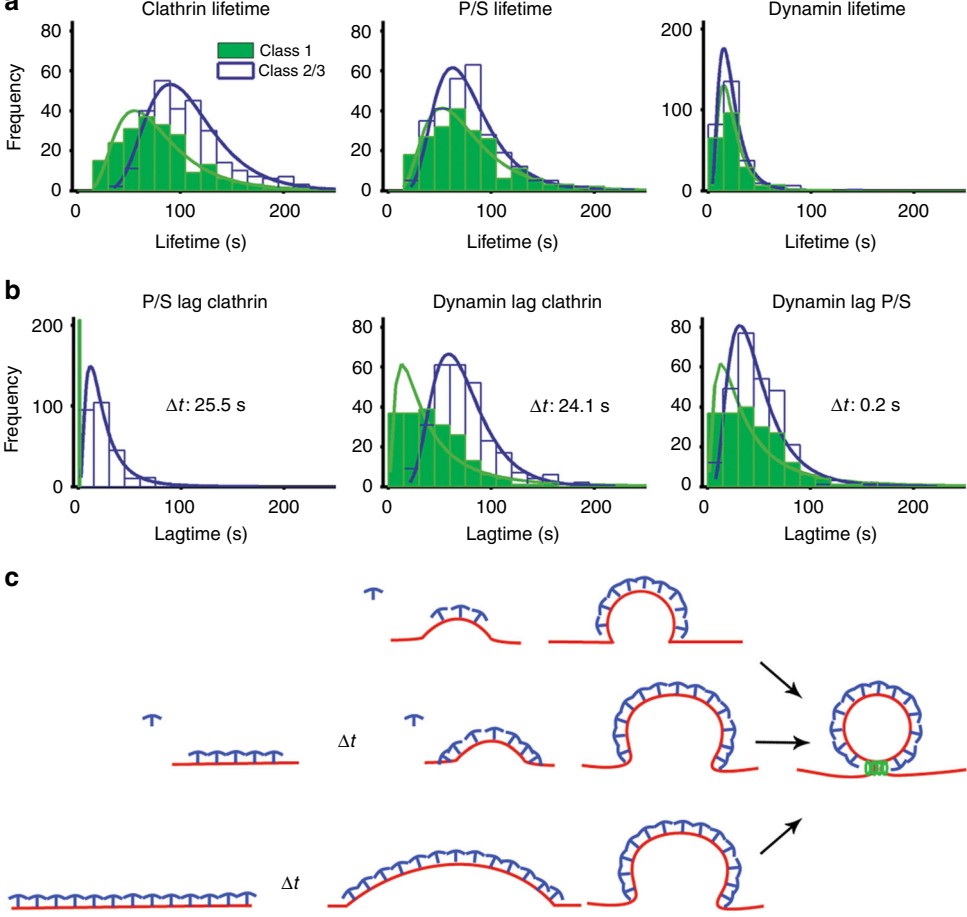

**Fig. 4** Lifetime analysis of clathrin events. **a** Lifetime distribution for clathrin, P/S, and dynamin; Class 1, green bars and line, Class 2/3 open with blue line. **b** Lag time for P/S and dynamin relative to the start of the clathrin events, and lag time for the start of dynamin relative to the start of P/S. **c** Relationship between clathrin assembly and membrane bending during CME

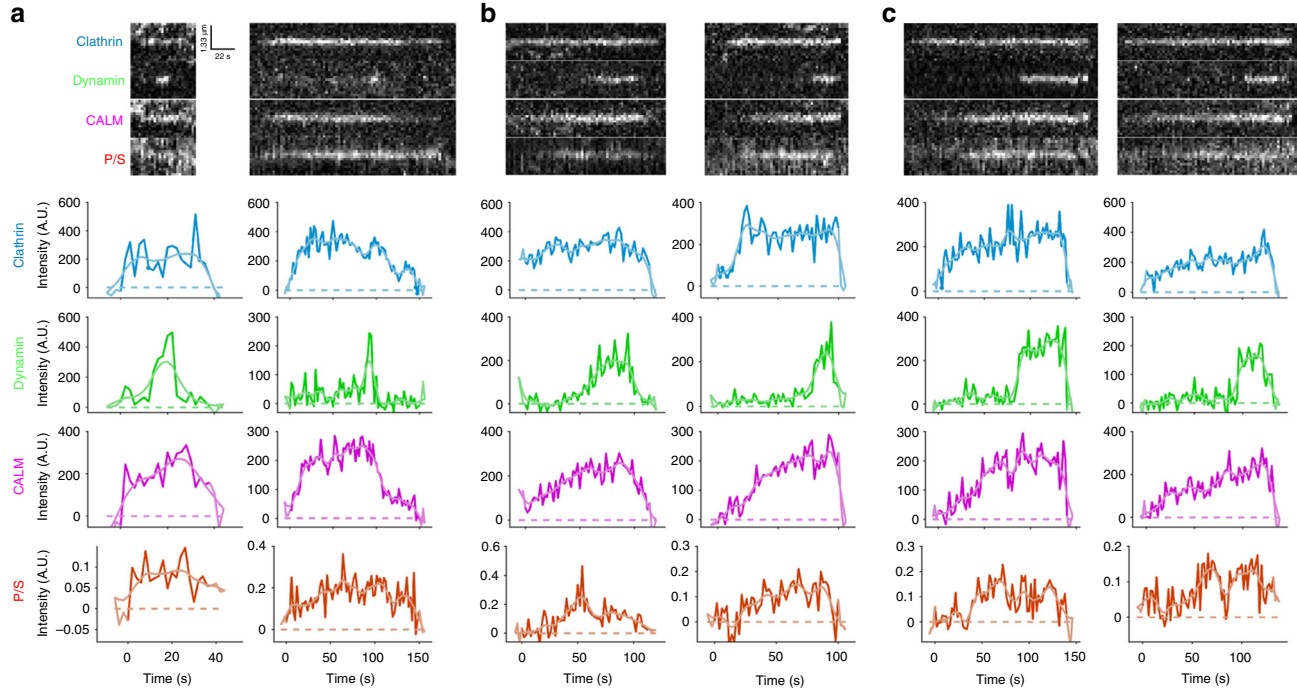

**Fig. 5** CALM recruitment is highly correlated with membrane bending. **a** pol-TIRF kymographs and intensity traces of clathrin, dynamin, CALM, and P/S are shown for Class 1 (**a**), Class 2 (**b**), and Class 3 (**c**)

## Methods

**Reagents.** 1,1′-Dioctadecyl-3,3,3′,3′-tetramethylindocarbocyanine perchlorate (DiI; Sigma, St. Louis, MO) was dissolved in DMSO to prepare 1 mg mL$^{-1}$ of stock. Cells were labeled using 1 µg mL$^{-1}$ DiI in 2.5% DMSO/HBSS (Hanks' Balanced Salt Solution, Cellgro, Corning). A volume of 100 µL DiI solution was added dropwise to 1 mL of HBSS and mixed by pipetting for <30 s, and subsequently washed 3 times in imaging buffer and visualized by pol-TIRF immediately for no more than 30 min. Imaging buffer was sterile Leibovitz's L-15 media with 10% (v/v) fetal bovine serum (FBS, Hyclone). The cells were imaged if p-polarized excitation intensities were <3000 to ensure that dynamics were not altered.

**Cell culture.** Human SK-MEL-2 cells (Parental SK-MEL-2 and hDNM2$^{EN}$) were a kind gift from D. Drubin (University of California, Berkeley, CA). The cells were cultured in DMEM (Hyclone, ThermoFisher), supplemented with 10% (v/v) FBS (Hyclone), penicillin and streptomycin, and glucose. For live-cell imaging experiments, the cells were plated on fibronectin-coated (Number 1.5) 25 mm coverslips (Thermo Fisher) at 50% confluency and imaged within 4–6 h of plating. Flame-cleaned coverslips were coated with fibronectin at a final concentration of 25 µg mL$^{-1}$ in dPBS for 30 min prior to plating cells. For visualization on the microscope, the coverslips were transferred to AttoFluor chambers (ThermoFisher) and were maintained in imaging buffer for up to 30 min.

**CRISPR donor vector design.** mTurquoise2[34], a gift from Dorus Gadella (Addgene plasmid # 60561), was amplified with BsaI sites on either end as well as KpnI and NotI recognition sequences (Supplementary Table 1). This amplicon was cloned into OCT4-EGFP-PGK-Puro[35], a gift from Rudolf Jaenisch Addgene# 31937, using KpnI and NotI restriction sites to create pDONOR2-mTq2. The P2A self-cleavable peptide sequence was cloned into pDONOR2-mTq2 in frame with the C terminus of mTurquoise2 using sequence and ligation-independent cloning. The gene encoding a variant of puromycin N-acetyl-transferase was amplified with primers containing NheI and NotI sites used for cloning in frame of the P2A sequence. This vector, pDONOR3-mTq2, was used for insertion of 1 kb CLTA homologous arms. Primers were designed containing BsaI recognition sequences with overhangs compatible to pDONOR3 and used to amplify regions adjacent to the CLTA stop codon (Supplementary Table 2). Arms were cloned into pDONOR3 by golden gate cloning.

**CRISPR generation of mTq2-tagged clathrin light chain in SK-MEL-2 cells.** A gRNA sequence for hCLTA adjacent to the stop codon of the open reading frame was chosen and cloned into Cas9 expressing vector pX330-U6-Chimeric_BB-CBh-hSpCas[36], a gift from Feng Zhang, addgene #42240 using BbsI sites and designated pCas9-hCLTA (Supplementary Table 3). pCas9-hCLTA and donor vector, pDO-NOR3-mTq2-hCLTA, were transfected into SK-MEL-2 enDyn-GFP using PEI in one well of a six-well plate. 72 h after transfection, the edited cells were selected using puromycin for 72 h. The single cells were then sorted into separate wells of a 96-well plate using a FACSJazz cell sorter. Clones were analyzed for insertion of mTq2 by PCR, sequencing, and western blot.

**Outside–outside PCR screening of clathrin-edited cells.** Primers specific to the CLTA locus outside of left and right homologous arms used in the donor vector were used for "outside–outside" PCR amplification. Phusion Hot Start II polymerase (Thermo Scientific) containing 2.5% DMSO and GC buffer was used for amplification of gDNA purified from the clone B1 and the parental unedited cells. PCR products were run on a 0.8% agarose gel containing gel red (Supplementary Table 4).

**Western blot.** Parental and gene-edited SK-MEL-2 cells were lysed in MPER buffer (Thermo Fisher) containing protease inhibitor cocktail (Sigma) on ice. Protein was quantified by Bradford assay and 20 µg of the sample was loaded per lane on a 10% SDS-PAGE gel. The proteins were transferred onto nitrocellulose membrane and blocked in PBS containing 0.1% Tween 20 and 3% BSA. After 30 min, Anti-clathrin H-55 antibody (Santa Cruz) and Anti-GFP B-2 antibody (Santa Cruz) were added at a 1:100 dilution to the membrane and incubated overnight. Blot was then stained using secondary antibodies conjugated to DyLight-680 and 700. Membrane was then imaged on a LICOR IR imaging system.

**CALM-iRFP cloning.** CALM-pmCherryN1[6] was a gift from Christien Merrifield (Addgene plasmid #27691) and piRFP682-N1[37] was a gift from Vladislav Ver-khusha (Addgene plasmid #45459). mCherry was replaced with iRFP682 using XmaI and NotI digestion and ligation. CALM-iRFP682 was amplified using the following primers and cloned into pIB2[38], a gift from Inder Verma (Addgene plasmid #12371), using AgeI and NotI digestion:

5′-GGGGTGGACCATCCTCTAGACTGCCGGAT CCACGCGTCCGTCAGA
TCCGCTAGCGCTA

3′-GATCTTCAATTGTTTTACGTATCTCGAATTCATGCATTTCACTCTTC
CATCACGCCGATC

**Retroviral transduction.** $1 \times 10^5$ 293T packaging cells were transfected for 48 h to generate retrovirus using 2.1 µg total DNA (1 µg pIB2-CALM-iRFP682, 1 µg pCL-

Eco, and 100 ng VSV-G[39], a gift from Bob Weinberg (Addgene plasmid #8454)) with 4 µg Polyethylenimine (linear, 25,000 g mol$^{-1}$, Polysciences, Inc., Warrington, PA). The supernatant was harvested and filtered through a sterile 0.3 µm filter, and added to $1 \times 10^5$ SK-MEL-2 for 48 h.

**TIRF-based fluorescence microscopy.** TIRF-based imaging was conducted using an inverted microscope built around a Till iMic (Till Photonics, Germany) equipped with a 60 × 1.49 N.A. oil immersion objective lens, diagrammed in Supplementary Figure 2. The microscope was enclosed in an environmental chamber and maintained at a temperature of 35–37 °C using heater fans. Excitation for pol-TIRF was provided by a 561 nm laser. Excitation for mTurquoise2 eGFP and iRFP682 constructs was provided by 445, 488, and 633 lasers, respectively. The 445, 488, and 561 nm lasers were combined into an acousto-optic-tunable filter and launched into a single-mode fiber. The 633 nm laser was launched into a separate single-mode fiber, and the beam paths were combined with a polychroic reflector ZT405/532/635rpc (2.2 mm substrate, Chroma Technology, Bellows Falls, VT). Laser excitation was sent to a 2D scan head (Yanus, Till Photonics), which, along with a galvanometric mirror pair (PolyTrope, Till Photonics), was used to position the laser focal spot in the back focal plane of the objective lens for 2-point and 360-TIRF illumination. A polychromic mirror reflected excitation wavelengths to the sample, ZT440/488/561/635rpc (3 mm substrate, Chroma Technology). Fluorescence emissions were first separated by a longpass dichroic mirror (ZT543rdc, 3 mm substrate, Chroma Technology) to reflect blue and green to one arm of the microscope, whereas a second longpass dichroic reflected blue to detector D4 (ZT488rdc, 3 mm substrate, Chroma Technology), and allowed green to pass to detector D3. Red and Far-Red fluorescence was transmitted to the second arm, and a longpass dichroic (T647lpxr, 2 mm substrate, Chroma Technology) allowed far-red fluorescence to pass to detector, D1, and reflected red to detector, D2. Bandpass filters were used in front of each detector (D4 (Tq2)—FF01-470/22, Semrock, Inc. Rochester, NY, D3 (eGFP)—FF02-510/10 m, Semrock, Inc. D2 (DiI)—ET595/50 m, Chroma Technology, and D1 (iRFP682)—FF01-680/42, Semrock, Inc.) and finally collected on four electron-multiplying charge-coupled device cameras (iXon3 885, Andor Technology, Belfast, Ireland). The magnified pixel size was 133 nm a side. The exposure time was held constant at 100 ms for 442, 488, and 633 excitations and 100–125 ms for P and S excitation. For 2-point images, the P and S images at each position were 50–62.5 ms. Laser powers were measured using a PM120D digital handheld power meter (ThorLabs, Newton, NJ) and were typically between 0.5 and 20 mW during imaging. Bias calibration was performed by acquiring a set of 30 images with a closed shutter in front of each camera. The image stack was averaged to calculate the representative bias level in each pixel. Bias images were subtracted from each frame in the raw data set.

**Back focal plane centering.** In order to determine the center of the objective lens' optical axis, a calibration was carried out daily and for each chamber used in data acquisition. This centering calibration ensured that the excitation laser light encountered the glass/cell interface with a single incidence angle and, hence, produced a single TIRF excitation volume. To map the angles of TIRF reflectance, a computer program steered the laser to positive and negative mirror positions, and thus incidence angles, over 1002 steps, and the intensity of the reflected light from the glass/water interface was read on a quadrant photodiode module. Intensity values were plotted as a function of mirror position and the half-maximal values were used to adjust the mirror angles to center the optical axis (Supplementary Figure 2).

**Fiducial data collection and image registration.** Images were registered using calibration images acquired simultaneously on each of the four EMCCD detectors. Briefly, 200 nm green beads (Life Technologies, Carlsbad, CA) immobilized on a glass coverslip were excited using 445 nm excitation, and the images were acquired as a single bead was moved across the field of view to create a well-sampled grid. The beads were localized in each channel, and a rigid affine transformation was used to transform all points onto the red channel.

**Simulation.** To predict the quantitative relationships between clathrin assembly and membrane-bending signals from pol-TIRF, we created a discrete 3D simulation in MATLAB (The MathWorks Inc, Natick, MA) and DIPimage toolbox version 2.8 (Delft University of Technology, Delft, The Netherlands), building on our previous work for 3D microscopy simulations[21,40,41]. The plasma membrane was represented as a plane that could either be bent into a sphere via a fixed radius of curvature (Fig. 1a, Supplementary Figure 1) or through progressive bending (Fig. 1b, Supplementary Figure 1), forming a vesicle of diameter = 100 nm.

For membrane bending during assembly, the forming clathrin pit was modeled as a sphere of fixed radius ($r = 50$ nm) intersecting a plane. By shifting the center of the sphere along the z direction, we obtain the topographies outlined in Supplementary Figure 1a. In this case, clathrin is assumed to cover the spherical cap and ultimately the spherical vesicle. In the case of clathrin assembly preceding curvature, the pit is modeled as a circular patch of membrane emanating from the plane of plasma membrane. Here the circular patch is designated to have uniform lateral clathrin intensity and the sphere is translated vertically over a progression of discrete radii. Thus, the vertical shift was set to $z_{shift} = \text{Area}/2\pi/r_i$, where $r_i$ ranged

from 25,000 nm (slightly bent) to 50 nm (fully formed sphere). This produced the progression in outlined in Supplementary Figure 1b.

The fluorophores were modeled relative to each discrete element of plasma membrane or clathrin coat. Since 360-TIRF illumination was used for the clathrin images, no orientation dependencies were modeled. Thus, the equation defining the 2-dimensional clathrin image is given by,

$$I_c(x, y, k) = N\left(\text{PSF}_{X,Y} \circledast \int_z I_c e^{-z/d} dz\right), \tag{1}$$

where, $I_c$ is the 3D $(x,y,z)$ distribution of clathrin during pit stage $k$, $d$ is the penetration depth of the TIRF field (100 nm), the microscope point spread function was modeled as a Gaussian distribution of full-width half max (FWHM) of 211 nm, typical of a 1.49 N.A. objective lens. Detection noise ($N$) was modeled by drawing intensities from a Poisson distribution.

Simulation of the pol-TIRF signals was achieved using the pol-TIRF fluorophore excitation equations of Axelrod and Anatharam[12] for the relative contributions of a plane and a sphere. Based on this work, we assume that the depth-dependent detection of emitted polarizations in the near field was approximately constant for a 1.49 N.A. objective, and could therefore be neglected. Thus, using our discrete model, the polarization for a growing pit could be described as a plane and spherical components excited by either p-pol or s-pol illumination.

$$I_p(x, y, z, k) = N\left(\text{PSF}_{X,Y} \circledast \int_z I_p^{sph} e^{-z/d} \sin^2\theta \sin^2\beta \right.$$
$$\left. + \cos^2\theta \cos^2\beta + I_p^{pla} \cos^2\beta dz\right) \tag{2}$$

$$I_s(x, y, z, k) = N\left(\text{PSF}_{X,Y} \circledast 1/2 \int_z I_s^{sph} e^{-z/d} \cos^2\theta \right.$$
$$\sin^2\phi \sin^2\beta + 2\sin^2\phi \sin^2\theta \cos^2\beta$$
$$\left. + \cos^2\phi \sin^2\beta + I_s^{pla} 1/2 \sin^2\beta dz\right) \tag{3}$$

Where, spherical coordinates $(\theta, \phi)$ are defined relative to the center of the sphere, $\beta$ is the angle between the dipole moment of DiI and the plane, $I^{pla}$ and $I^{sph}$ are the intensities/unit membrane in the plane and sphere (assumed equal). Beta was determined by measuring planar regions of the plasma membrane and measuring the regional minimum that was found to be 0.26, which sets $\beta=70°$, which is nearly identical to the 69° value measured by Anantharam et al.[12].

Discrete simulations were conducted on a 2 nm grid over pit morphological states, and then downsampled to nominal microscope pixel dimensions of 125 nm in $x$ and $y$, and blurred with a Gaussian PSF with FWHM = 211 nm. Detection noise ($N$) was modeled by drawing intensities from a Poisson distribution. The code for this simulation will be made available via the Mathworks File Exchange.

**Correlative pol-TIRF–TEM and pol-TIRF–AFM.** SK-MEL-2 expressing endogenous clathrin-Tq2 were plated on fibronectin-coated coverslips for 5 h, labeled with DiI, and sonicated with a Branson Digital Sonifier 450 in stabilization buffer (70 mM KCl, 30 mM Hepes, 5 mM MgCl₂, at pH 4, and 1 mM DTT) with a 1/8" tapered microtip ~5 mm above the coverslip for a single 400 ms pulse at 10% amplitude. Stabilization buffer was immediately removed after sonication using an aspirator and fixed with 2% formaldehyde, para (PFA) (Fisher Scientific) for 20 min followed by pol-TIRF imaging in PBS. In order to identify the same cells for correlative EM, a 9 × 9 grid of fluorescent images surrounding the cell of interest was collected, and a circle was drawn on the underside of the coverslip approximately around the imaged area using a high precision fine diamond scriber with a 0.5 mm diameter tip (Electron Microscopy Sciences). The coverslips were mounted on a slide with 10 µL of 2% glutaraldehyde to keep the sample hydrated (Sigma Aldrich). The coverslip was sealed with VALAP (1:1:1 mixture of Vaseline, lanolin, and paraffin) and epoxy prior to shipment for TEM or AFM.

**Electron microscopy.** Coverslips were transferred from glutaraldehyde into freshly prepared 0.1% w/v tannic acid in water and incubated at room temperature for 20 min. They were then rinsed 4× in water and transferred into 0.1% w/v uranyl acetate and incubated for 20 min, and rinsed with water prior to dehydration. Dehydration into ethanol, critical point drying, coating with platinum and carbon, replica lifting, and TEM were performed as previously described[42]. Replicas were placed onto Formvar/carbon-coated 75-mesh copper TEM grids (Ted Pella 01802-F).

**Atomic force microscopy.** Correlated fluorescence-AFM was performed using an Olympus IX73 inverted microscope equipped with an Olympus PlanApo 60 × 1.45 N.A. oil immersion objective and Hamamatsu ORCA-Flash 4.0 V2 CMOS camera. The system was integrated with an Asylum Research MFP-3D-BIO AFM system and placed in a vibrational isolation chamber. AFM scanning of the unroofed cells in PBS solution was conducted at room temperature, under non-contact/AC mode, using Mikromasch CSC381/Cr-Au cantilevers, with a nominal spring constant of 50 pN nm⁻¹ and a resonance frequency of 14 kHz. AFM image analysis was

performed in Asylum Research AFM software. The correlative fluorescence-AFM images were aligned based on the correlation of spots between the two sets of clathrin fluorescence images to define fiducial markers for image registration, and quantified as described below.

**Image correlation.** Clathrin structures and positions were manually identified in the EM images by the appearance of the honeycomb lattice. The radius and centroid of each object was manually determined by fitting circles on the clathrin structures until it was completely encompassed. The ultrastructures were qualitatively categorized as [1]—flat, [2]—shallow curvature, [3]—medium curvature, [4]—domed curvature, and [5]—formed clathrin vesicle, based on the relative shadowing on the edge of the object. Finally, the coordinates of Tq2-clathrin were obtained by fitting the spots to a 2D Gaussian, and identifying at least four spots to use as fiducial markers to generate the 2D affine transformation matrix that minimized the distance between correlated spots.

**Tilt-series tomography image analysis.** The radius and heights of clathrin ultrastructures were quantified by manually fitting circles to the $xy$-sum projection to determine the radius, and drawing an arc along the $yz$-sum projection of the tomogram to determine the height.

**Correlative dSTORM-platinum replica transmission electron microscopy.** SK-MEL-2 cells for correlative microscopy in Fig. 1 were obtained from ATCC and were grown in DMEM lacking phenol red (Life Technologies 31053-036) and were supplemented with 10% v/v FBS with 1% v/v Glutamax (Life Technologies 35050-061), 1% v/v Penicillin/Streptomycin (Invitrogen 15070-063), and 1 mM sodium pyruvate (Sigma S8636). The cells were transfected with EGFP-clathrin light chain (a) using lipofectamine 2000 on day 1. On day 2, they were sorted to obtain only GFP-containing cells and plated on coverslips embedded with gold nanoparticles (hestzig.com, part #600-200AuF). The coverslips had been coated with a 1:40 solution of fibronectin (Sigma F1141) in PBS for 30 min. On day 3, the cells were unroofed and labeled as described below.

First, the coverslips were rinsed in stabilization buffer (70 mM KCl, 30 mM HEPES brought to pH 7.4 with KOH, 5 mM MgCl₂) for 2 min and unroofed by sonication in 2% paraformaldehyde (PFA) in stabilization buffer. They were then fixed in 2% PFA for 20 min. After rinsing with PBS, the cells were placed in a blocking buffer (3% bovine serum albumin in PBS) for one hour. They were then immunolabeled with 11 nM Alexa Fluor 647-labeled GFP-nanotrap (preparation described below) in blocking buffer for 45 min, rinsed in PBS, and post fixed in 2% PFA for 20 min. The coverslips were then imaged in a sealed chamber containing blinking buffer (10% w/v glucose, 0.8 mg mL⁻¹ glucose oxidase, 0.04 mg mL⁻¹ catalase, and 100 mM 2-mercaptoethanol made fresh in PBS immediately before imaging). dSTORM was performed on a Nikon NSTORM system with 10 kW cm⁻² 647 nm laser in TIRF illumination with 30,000 10 ms frames. A final image was created with Nikon Elements NSTORM analysis software with 5 nm pixel spacing.

After imaging, the coverslips were marked with a diamond objective marker (Leica 11505059). The oil was cleaned off of the coverslip with 80% ethanol. They were then stored in 2% glutaraldehyde in PBS and processed for EM the following day. EM processing and imaging was performed as described above. The gold nanoparticles that were embedded in the coverslips were visible in both dSTORM and EM, and were therefore used as spatial fiducial markers. Three gold nanoparticles were used to map the fluorescence onto the EM image using an affine spatial transformation and nearest neighbor interpolation.

GFP-nanotrap was expressed and purified as previously described[43]. It was then labeled with Alexa Fluor 647 NHS ester (ThermoFisher 37573) using 2.4 molecules of dye for every one nanobody. These were purified using size exclusion chromatography and concentrated to 11 µM. SDS-PAGE indicated 1–4 dyes per nanobody. All GFP-nanotrap concentrations were estimated assuming an $A_{280}$ extinction coefficient of 26,930 M⁻¹ cm⁻¹.

**Data availability.** The data sets generated for this current study are available upon request from the corresponding author.

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

## Acknowledgements

This material is based on the work supported by the National Science Foundation under CAREER Award to ADH, Grant No. 0953561, the National Science Foundation/EPSCoR Cooperative Agreement #IIA-1355423, the South Dakota Research and Innovation Center, BioSNTR, and by the State of South Dakota BOR CRGP to ADH and DMR-1337586 and DMR-1206908 to S.S. Any opinions, findings, and conclusions or recommendations expressed in this material are those of the author(s) and do not necessarily reflect the views of the National Science Foundation. K.A.S. and J.W.T. were supported by the Intramural Research Program of the National Heart Lung and Blood Institute, National Institutes of Health. We thank the NHLBI, NIH electron microscopy and light microscopy core facilities for use of equipment. We thank Alan Hoofring of NIH Medical Arts for art in Figs. 1a, b and 2a.

## Author contributions

A.D.H., B.L.S., K.A.S., S.S., and J.W.T. designed the study and interpreted the data. B.L.S., K.A.S., S.T.L.-N., E.M.B., Q.L., A.H., A.M.D., and J.G.K. performed the experiments, analyzed the data, and interpreted the data. A.D.H., S.S., and J.W.T. supervised the project.

## Additional information

**Competing interests:** The authors declare no competing financial interests.

