## [Peer Review File · Nature Communications]

Reviewer #1 (Remarks to the Author):

All my previous questions and concerns have been addressed and I thus recommend publication of the paper in Nat. Communications.

Reviewer #2 (Remarks to the Author):

This study represents one of the most technically impressive collections of data focused on a single, highly controversial question: can clathrin accommodate multiple membrane bending modes? In my opinion, the authors have finally answered this incredibly challenging question in the affirmative, providing a convincing description of clathrin-mediated endocytosis that should help to unify the field. The issue now falls to the question of whether or not a careful description of this process is a sufficient advance, or whether functional data beyond localization studies are necessary to better understand what actually drives membrane bending. The authors indicate that "a complete mapping of all possible proteins involved in this process and subsequent changes in these factors during signaling states is beyond the scope of this current work." I entirely agree, but I don't believe any of the previous reviewers asked for such an in-depth analysis. Instead, some simple depletion experiments would likely be sufficient to understand the contributions of key endocytic adaptor proteins. However, the authors seems reticent to address this shortcoming. Ultimately, this manuscript should without a doubt be published, but it would be even more compelling with some relatively modest functional studies, perhaps focused on early-acting adaptors that have previously been suggested to contribute to initial membrane bending events

Reviewer response to “Membrane bending begins at any stage of clathrin-coat assembly and defines endocytic dynamics.”

Reviewer #1 (Remarks to the Author):

All my previous questions and concerns have been addressed and I thus recommend publication of the paper in Nat. Communications.

Reviewer #2 (Remarks to the Author):

This study represents one of the most technically impressive collections of data focused on a single, highly controversial question: can clathrin accommodate multiple membrane bending modes? In my opinion, the authors have finally answered this incredibly challenging question in the affirmative, providing a convincing description of clathrin-mediated endocytosis that should help to unify the field. The issue now falls to the question of whether or not a careful description of this process is a sufficient advance, or whether functional data beyond localization studies are necessary to better understand what actually drives membrane bending. The authors indicate that "a complete mapping of all possible proteins involved in this process and subsequent changes in these factors during signaling states is beyond the scope of this current work." I entirely agree, but I don't believe any of the previous reviewers asked for such an in-depth analysis. Instead, some simple depletion experiments would likely be sufficient to understand the contributions of key endocytic adaptor proteins. However, the authors seems reticent to address this shortcoming. Ultimately, this manuscript should without a doubt be published, but it would be even more compelling with some relatively modest functional studies, perhaps focused on early-acting adaptors that have previously been suggested to contribute to initial membrane bending events.

Response: We thank this reviewer for recognizing the value of the work and we do agree that the analysis of membrane bending during genetic manipulation of CME machinery should prove to be highly informative. We anticipate addressing this in future studies as the postdoc carrying out this work has moved to a new lab.